# Molecular Mechanisms of Silicone Network Formation: Bridging Scales from Curing Reactions to Percolation and Entanglement Analyses

**DOI:** 10.3390/polym17192619

**Published:** 2025-09-27

**Authors:** Pascal Puhlmann, Dirk Zahn

**Affiliations:** Lehrstuhl für Theoretische Chemie/Computer Chemie Centrum, Friedrich-Alexander Universität Erlangen-Nürnberg, Nägelsbachstraße 25, 91052 Erlangen, Germany

**Keywords:** silicone, polymerization mechanisms, molecular dynamics, network analyses

## Abstract

The curing of silicone networks from dimethylsilanediol and methylsilanetriol chainbuilder–crosslinker precursor mixtures is investigated from combined quantum/molecular mechanics simulations. Upon screening different crosslinker content from 5 to 15%, we provide a series of atomic-resolution bulk models all featuring 98–99% curing degree, albeit at rather different arrangement of the chains and nodes, respectively. To elucidate the nm scale alignment of the polymer networks, we bridge scales from atomic simulation cells to graph theory and demonstrate the analyses of 3-dimensional percolation of -O-Si-O- bonds, polydimethylsiloxane branching characteristics and the interpenetration of loops. Our findings are discussed in the context of the available experimental data to relate heat of formation, curing degree and elastic properties to the molecular scale structural details—thus promoting the in-depth understanding of silicone resins.

## 1. Introduction

Silicones are widely spread in both industry and everyday life, with applications ranging from lubricants, glues, and sealants to flexible bulk materials [1,2]. The cured polymers are comparably non-reactive, considered as practically non-toxic and may be formed from inexpensive constituents. Using dimethylsilanediol (DMS-D) as base material for divalent linking, polydimethylsiloxane (PDMS) chains are simply obtained by condensation reactions. In turn, curing of mixtures of DMS-D and the trivalent species monomethylsilanetriol (MMS-T) leads to the formation of branched chains and crosslinked polymers, which network alignment may be tuned from precursor chemistry, temperature, spatial confinements, and most obviously by tuning the ratio of the constituents [3]. Based on decades of experimental evidence, industrial application benefits from a wealth of experience for engineering mechanical and thermal properties of PDMS based polymer materials [4,5,6].

The properties of solid silicones are thus tailored by (i) the degree of crosslinking within polymer networks, (ii) the type of crosslinks, and (iii) the nature of the strands between the crosslinks. While all these features correlate with each other, the former two aspects are mainly defined by chemical reactions, whereas the latter reflect geometric confinements and local entanglement. To unravel the origin of silicone materials properties from modeling and simulation, suitable approaches thus need to bridge from the quantum chemistry of condensation reactions, the molecular mechanics of polymer strands and crosslinks to the percolation and branching characteristics of the overall network, respectively. Molecular dynamics simulations combined with QM/MM assessment of condensation reactions proved a valuable tool for curing polymers, albeit so far focusing on linear macromolecules (such as silicone oil based on PDMS [7]) or bulk epoxy resins that feature particularly high levels of crosslinking [8,9], respectively. The soft nature of solid silicone, however, stems from the subtle interplay of flexible strands and a moderate number of crosslinks that basically tune the mechanical properties between two rather unlike limiting cases: lubricant silicone oil and brittle silica ceramics.

The importance of network topology motivates the use of graph theory [10,11,12,13] for interpreting crosslinked polymers [14,15,16,17,18,19,20]. In what follows, we extend our QM/MM based molecular dynamics simulation approach to the curing of DMS-D/MMS-T mixtures to create unbiased models of crosslinked PDMS networks at atomic resolution [9]. While assessing heat of formation, curing degree and elastic properties at atomic level of accuracy, we furthermore coarsen our structures by graph theory to interpret the materials properties in terms of knot and percolation analyses, respectively.

## 2. Model and Methods

The MD simulations were carried out using LAMMPS package (March 2023 version), using an integration time step of 1 fs [21]. To maintain constant pressure (1 bar) and temperature (300 K), the Nosè–Hoover algorithm was employed, with relaxation times of 1 ps and 0.1 ps, respectively [22]. This setup of the MD simulations is adopted from our earlier study of linear PDMS formation [7]. Likewise, the molecular mechanics force field for describing the interactions of Si, O, OH, and methyl species is adopted from that previous study of silicone oil. Likewise, a real-space cutoff distance of 12 Å was employed for evaluating the real-space part of all pairwise potentials, whereas long-range electrostatic interactions were treated using the particle mesh Ewald summation method [23].

While the employed force fields allow for Si-O bond dissociation and reorganization, QM considerations are needed to implement the condensation reactions of -Si-OH moieties. In full analogy to ref. [7], we use a nearest-neighbor OH-OH distance criterion for assessing candidate reaction sites. Upon sampling the energy difference in our model system before/after a possible reaction event, the reaction energy Δ*H* is calculated using(1)$$∆H=\langle E_{total}\left(-Si-O-Si-\right)\rangle -\langle E_{total}\left(-Si-OH\ldots OH-Si-\right)\rangle -0.023\;eV$$
where *E_total_* reflects the system energy from molecular mechanics before/after the condensation reaction and the additional −0.023 eV energy term accounts for the energy of the proton transfer as assessed from QM calculations [7]. We dismiss endothermic reaction steps and in turn implement exothermic reaction steps by removing the product water from the simulation cell [7]. Within increasing degree of crosslinking, this calls for careful sampling as the stiffening of the polymer network reduces the pace of structural relaxation quite drastically.

For this reason, an adaptive sampling protocol was developed. Upon attempting the formation of a new Si-O-Si bond, we find 1 ps of relaxation is needed for rearranging the involved O^2−^ ion position (from the pristine Si-OH-Si arrangement of the reactant state). Next, the heat of reaction is sampled from sketches of 10 ps MD runs in both reactant and product state. From comparing the averages sampled from the first 5 ps and the following 5 ps runs, respectively, we provide an error estimate in assessing the reaction energy Δ*H*. In case of deviations by more than 10%, we increase the relaxation runs by additional 10 ps—and thus allowing further relaxation of the polymer network. The assessment of Δ*H* is then repeated iteratively. An illustration of DMS-D/MMS-T condensation reactions and silicone resin formation is provided in Figure 1.

All MD simulations of the polymerization of DMS-D/MMS-T mixtures start from 500 unreacted monomers which were randomly placed in a cubic cell. The initial cell dimensions of 80 Å were chosen generously large to avoid molecular overlap. Next, periodic boundary conditions are applied to mimic a bulk precursor liquid—which was pre-equilibrated by 2 ns at ambient conditions before implementing the condensation reactions. In parallel setups of each 5 independent runs, we explore 85/15, 90/10 and 95/5 mixtures of DMS-D/MMS-T species, respectively. For data analyses and visualization Python 3.12, NetworkX 3.3 and VMD 2.0 was used [24,25].

To analyze the overall polymer network, we coarsen our atomic models by means of graph theory. For this, we focus on the silicone backbone formed by silicon and oxide ions—thus ignoring the lateral methyl groups and unreacted hydroxide moieties, respectively. A contact matrix of all chemically bonded silicone building blocks is created by means of a Si-O distance delimiter. For this purpose, a cutoff of 1.8 Å was found suitable, thus allowing up to 0.2 Å elongation from the Si-O equilibrium distance of 1.60 Å. The contact matrix is converted into a graph by interpreting the Si atoms as “nodes”, whereas the “edges” represent oxide ions acting as connections between the nodes. To each node, we attribute the number of directly bonded nodes as the degree of connectivity, thus discriminating terminal nodes, linear building blocks, and crosslinking of the silicone strands, respectively [26]. Subject to the degree of crosslinking in the network, the overall graph may consist of a number of subgraphs of which each subgraph represents a silicone strand. To this end, two silicone strands that are not connected via chemical bonding are considered as separate molecules and thus separate subgraphs, respectively.

In such graph theoretical representation, the spatial arrangement of the strands is ignored and we instead only focus on connectivity. On this basis, a stand-alone PDMS strand is represented by a simple line—which length reflects the number of Si atoms involved. In turn, crosslinks are represented by additional lines starting out from the node mimicking the underlying MMS-T moiety. Moreover, we identify loops by avoiding virtually ‘infinite’ silicone strands of periodically re-occurring nodes. In graph theory, loops are represented as circles, thus ignoring the spatial arrangement within the atomic simulation model. To this end, the definition of loops is independent from the location of the box boundaries of the simulation cell. However, for further characterization, we discriminate closed loops within the simulation cell from percolating strands, that is node–node contacts crossing the cell boundaries of our periodic simulation systems [27,28]. As there may be a multitude of percolating polymer strands, graph theory is furthermore used to distinguish independent percolations, i.e., separate circles, from strands featuring common notes. To avoid double-counting, focusing on the shortest circle among nestled circles thus offers to diminish ‘detours’ when studying percolating strands. As a consequence, our graph theory analysis enables the identification of independent bundles of polymer strands that percolate the simulation cell.

To assess the elastic properties of the finally obtained polymer systems, we performed MD runs at 300 K and 1 atm using the anisotropic implementation of the barostat algorithm [22]. To assess the Youngs moduli Y_ii_ (with i = x,y,z), the models were scaled to 1% elongation in the corresponding direction whilst maintaining volume by deformation in the perpendicular directions. For reliable sampling of stress tensor averages, we then performed NVT simulations at 300 K. A relaxation period of 100 ps was found appropriate before sampling stress averages from additional 300 ps runs. To estimate the error margins of the sampling process, we furthermore split our statistics in two halves (0–150 ps and 150–300 ps, respectively) and assume the deviation as the error margin of the overall average. On this basis, the statistical error of sampling the Youngs moduli at 300 K was found <5% for all systems investigated.

To assess the glass transition temperatures T_g_, we performed heating/cooling runs between 50 K and 500 K, respectively, at rates of 10 K/ns. From this, we analyzed profiles of the system density as functions of temperature. Upon fitting linear asymptotes to the profiles obtained by heating and cooling, the resulting crossing points are taken as estimates for T_g_, respectively. This procedure was adopted from our earlier work on epoxy resins reported in Ref. [9].

## 3. Results

The intrinsic complexity of weakly ordered polymer networks poses the challenge of avoiding artificial biasing from ad hoc assumptions when creating the underlying simulation models. We therefore employ our simulation protocols to study polymer formation from the very beginning, namely using liquid mixtures of the precursors as starting points. In the liquid state, any bias from initial placement of the precursors may be amended from relaxation runs using MD simulations at constant temperature and pressure. For this purpose, we find 2 ns equilibration runs sufficient for providing unbiased liquid phase models of the precursor mixtures at ambient conditions.

We then implemented condensation reactions in our MD studies following our hybrid QM/MM protocol reported in ref. [7]. Thus, nearest-neighbor contacts of candidate hydroxy groups are identified and subjected to reaction attempts. The assessment of the underlying reaction energy according to Equation (1) calls for reliable sampling of the molecular mechanics energy terms before and after the reaction attempt. At the early stages of curing, we can fully adopt the sampling protocols from our earlier study of silicone oil polymerization [7]. However, upon solidification of the present silicone models, we encounter slow relaxation modes resulting from the change in viscosity experienced by the simulation systems. For this reason, the adaptive sampling scheme described in the Section 2 becomes indispensable to ensure reliable assessment of the heat of reaction. In terms of network formation, our protocol needs to discriminate exothermic condensation reactions from endothermic attempts (which are dismissed). While this in principle reduces to a yes/no decision tree, we nevertheless aim for proper assessment of the overall heat of polymer curing. For this purpose, the strict sampling criterion described in Section 2 is needed to impose the desired 10% error margins for the calculation of reaction energy at reasonable accuracy.

A typical curing process based on our simulation scheme is illustrated in Figure 1. Based on nearest-neighbor contacts, exothermic condensation reactions are implemented in an iterative manner. The initial mixture of DMS-D and MMS-T monomers quickly evolves into dimers, trimers, etc., with practically all reaction attempts being accepted as exothermic polymerization steps. However, upon increasing degree of polymer curing, we find the number of failed reaction attempts to increase exponentially. Our iterative protocol for attempting/implementing further condensation reactions is therefore stopped upon reaching a maximum number of failed attempts. As termination criterion, we suggest a total of 10^5^ reaction attempts—which is 200 times more than the theoretical minimum of 500 polymerization steps required for the curing of 500 DMS-D monomers.

For each composition (85/15, 90/10 and 95/5) of the DMS-D/MMS-T mixtures we studied, 5 parallel curing runs were performed, respectively, to elucidate the statistical significance of modeling the curing processes. Profiles of the curing process as a function of the number of condensation reaction attempts are illustrated in Figure 2. Both, the overall shape of the curing profiles and the finally achieved curing degree are within <1% margins, thus suggesting that our models of 500 monomers are well suited for studying silicone network formation. Likewise, also our sampling of the overall heat of formation and the mass density of the finally obtained silicone polymer is found within reasonably small deviations (Table 1). This data nicely complies with the experimental data available for 98–99% curing of PDMS networks exhibiting mass density and heat of formation within the given error margins [4].

In turn, our simulation data shows surprisingly large error margins for the assessment of elastic properties. While silicone elastomers show Youngs moduli in the range of 1–100 MPa [6,29,30], our systems clearly reflect silicone resins—for which experimental reference is scare, but available data on phenylsilicone at least qualitatively confirms the 2–3 GPa scale observed for the averages taken from our simulation models [31]. However, when comparing the results from the 5 independent simulation runs performed for each precursor mixture, we already find root-mean-square deviations in the ballpark of 10–25% for both the Youngs and bulk moduli, respectively. Moreover, when scrutinizing the Youngs modulus *Y* = <*Y*_xx_, *Y*_yy_, *Y*_zz_> into the elastic responses in 3 different directions (x, y, z), we find considerable deviations even within an individual simulation model. This hints at serious limitations regarding the assessment of the spatial arrangement of the polymer network. While the computational demand for QM/MM-based silicone curing simulations from 500 precursor species is already quite large, we argue that even larger models are needed to obtain spatial averages at more desirable error margins.

To expedite on the relation of network structure and elastic properties, we may however benefit from the small size of our simulation cells. Indeed, each model is subjected to periodic boundary conditions and thus reflects a quasi-crystal of ~40 Å unit cell dimensions. So rather than attempting to average out spatial inhomogeneity, in what follows we characterize the individual polymer networks and try to interpret the observed elastic properties in terms of the underlying structures.

To this end, we suggest graph theory for elucidating the connectivity of the polymer network at the 40 Å scale. Coarsening the -O-Si-O- backbone into graphs, DMS-D silicone strands are reduced to lines of different length, whereas branching via MMS-T moieties is shown as nodes connecting several lines. An illustration for this classification is provided in Figure 3a. The only exception to this line/node classification is given by self-connecting strands—which reflect polymer loops. These may exist as entities within the simulation cell (real loops) or in terms of head-to-tail connections that arise from the periodic boundaries of our simulation setup. The latter situation is key to identifying percolations, that is continuous -O-Si-O- covalent bonding throughout the bulk material (see Figure 3b for illustration). A percolation is hence defined as a strand spanning from one site to the opposite site of the periodic simulation box [27]. By counting the number of percolating strands that connect via the box boundaries in x-, y- and z-direction (and normalization with respect to the area of the box face), we furthermore define a ‘percolation density’ in the corresponding direction, respectively. We argue that percolation density and the length of the percolating strands should be of relevance for describing the structure and mechanic properties of resins—that is chemical bonding throughout the bulk polymer. In turn, we suggest the number of interpenetrating loops (see Figure 3c for illustration) for describing thermosets featuring physical bonding rather than percolation of covalent bonding.

On this basis, we clearly identify the importance of -O-Si-O- bond percolation for all of the DMS-D/MMS-T mixtures we studied. Indeed, we always find at least one percolation in any direction, typically more (Table 2). In turn, the co-existence of loops was only very rarely observed (for the system shown in Figure 3). However, since also this system features percolation of the covalent bonding in all directions, we argue that our silicone models should be generally considered as chemically bonded resin networks rather than interpenetrated loops.

While this overall character is fully in line with the observed 2–3 GPa scale Youngs moduli, attempts to scrutinize the relations between elastic properties and percolation density showed only moderate success. In Figure 4, we plotted the observed Youngs moduli as functions of percolation density. To some extent, a linear trend is observed, with the Youngs modulus increasing from 2 to 3 GPa upon comparing the lowest and largest percolation densities of our silicone resin models. Moreover, we explored the role of the length of the shortest percolating -O-Si-O- strand. Arguably, shorter strands may not evade Si-O bond elongation by the straightening of curved sketches such that we would expect particularly large restoring forces upon tensile testing. In turn, longer and more curvy strands only need to perform bending of O-Si-O and Si-O-Si valence bonds—albeit at the price of larger scale atomic displacements within the bulk solid. While some correlation is observed, namely a trend to low Youngs moduli for the systems featuring 5% MMS-T content, we caution that the entire 2–3 GPa range of the Youngs moduli is readily present for the short strands at 10–15% MMS-T content, respectively (Figure 4).

To account for the complexity of the 3-dimensional polymer network, our analyses were performed for both directionality-correlated datasets and averages taken over the data collected for the x-, y- and z-directions, respectively. From this, we find that the weak correlations as discussed above only apply to directionally linked datasets—such as *Y*_xx_ plotted as a function of percolation density in x direction—whereas averages including the perpendicular directions largely diminishes any attempt to relate mechanical properties to network structure.

## 4. Conclusions

We demonstrated the investigation of silicone resins by means of unbiased molecular simulations models. This encompasses the entire formation process, that is the curing of precursor mixtures into 3-dimensionally percolating polymer networks. Subject to the composition of the DMMS-D/MMS-T precursor mixture, we find near 99% polymerization upon formation of 0–3 residual molecules of finite length. The mechanical properties are determined by a single, covalently bonded -O-Si-O- network—which was found to percolate throughout the 4 nm scale dimensions of our simulation cells with periodic boundary conditions. We point out that our models do not reflect silicone elastomers (which are comparably soft), but to densely crosslinked resins—for which we indeed find quite good agreement with the available experimental data.

Using graph theory, we condensed the network analyses to the interpretation of branched strands and finite loops, respectively. The latter reflect interpenetrating loops and thus feature ‘physical bonding’, whereas percolating strands of covalently bonded -O-Si-O- strands appear to dominate the elastic properties such as Youngs moduli in the range of 2–3 GPa, respectively. Relating the structural analyses of the polymer networks to the observed elastic moduli, we identified two characteristics that help to interpret the stiffness of the silicone resins studied. The elaborated structural descriptors are based on the number density of percolating polymer strands and the length of the shortest series of -O-Si-O- bonds connecting the simulation boxes via the periodic boundaries. Based on our molecular dynamics simulations, both of these descriptors show significant variation when inspecting nm scale subsets of the polymer network. In turn, we conclude that 3-dimensionally homogeneous polymer networks call for much larger simulation cells, ideally exceeding the named 4 nm scale dimensions by at least an order of magnitude.

## Data Availability

The original contributions presented in the study are included in the article, further inquiries can be directed to the corresponding author.

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
