# Peer review of "Molecular Mechanisms of Silicone Network Formation: Bridging Scales from Curing Reactions to Percolation and Entanglement Analyses"

_polymers, 2025, doi:10.3390/polym17192619_

Round 1
Reviewer 1 Report
Comments and Suggestions for Authors
This work employed molecular dynamics techniques to investigate the curing behavior and properties of silicone, but this manuscript lacks complementary experiments. In addition, there are some issues with this manuscript that need to be improved. The suggestions are as follows:
- In the abstract section, it is necessary to highlight the reason for conducting this research, and at the end, the significance of this research for future related work should also be clearly stated.
- Are there any relevant reports on the research of silicone curing behavior using molecular dynamics and the prediction of its performance? The introduction should list relevant previous studies and highlight the innovativeness of this work.
- 3. The table should be in three-line format.
- As can be seen from Table 1, the different proportions of the samples are obvious. Why are the obtained curing degree values basically similar?
- It is reasonable to predict the performance of materials using molecular dynamics, but there is a lack of comparison between experimental data and simulated data in the manuscript?
- The manuscript lacks the molecular structure of the simulated monomer, the structure of the curing agent, and the schematic diagrams of their curing processes.
Author Response
Reviewer 1
Comments and Suggestions for Authors
This work employed molecular dynamics techniques to investigate the curing behavior and properties of silicone, but this manuscript lacks complementary experiments. In addition, there are some issues with this manuscript that need to be improved. The suggestions are as follows:
- In the abstract section, it is necessary to highlight the reason for conducting this research, and at the end, the significance of this research for future related work should also be clearly stated.
Our mission to promoting the in-depth understanding of silicone resins is now pointed out more clearly.
- Are there any relevant reports on the research of silicone curing behavior using molecular dynamics and the prediction of its performance? The introduction should list relevant previous studies and highlight the innovativeness of this work.
To a large extent, this specific field of using QM/MM based MD simulations was pioneered in refs. 7 and 9. The relevant literature is discussed there.
- The table should be in three-line format.
Done.
- As can be seen from Table 1, the different proportions of the samples are obvious. Why are the obtained curing degree values basically similar?
The curing degree is based on hundreds of reaction events, i.e. reflecting excellent statistics with small fluctuations. In turn, our 5 nm scale simulation cells comprise only a few percolating silicone polymer strands – thus giving rise to different proportions. Larger cells would help to diminish also those fluctuations. However, as we pointed out at the beginning of the graph theory analyses, in the present work we purposefully use 1-3 nm scale inhomogeneities of silicone resins to outline relations of structure and elastic properties.
- It is reasonable to predict the performance of materials using molecular dynamics, but there is a lack of comparison between experimental data and simulated data in the manuscript?
We did provide all available comparison by evaluating density, glass transition temperature and elastic moduli. See table 1 and related discussion.
- The manuscript lacks the molecular structure of the simulated monomer, the structure of the curing agent, and the schematic diagrams of their curing processes.
An illustration of DMS-D/MMS-T condensation reactions and silicone resin formation is provided in scheme 1.

Reviewer 2 Report
Comments and Suggestions for Authors This manuscript is about the numerical simulation of silicon networks. The authors investigate the relations between system preparation, network structure and mechanical properties. They use a previously published mixed QM-MM molecular model for describing the crosslinking process. The network analysis is described in terms of percolating "strands", loops and free molecules. The main point of the manuscript is, that percolation of strands are the more likely to correlate to the Young Modulus. Due to the limited size of the simulation box (QM-MM remains challenging), the mechanical results are quite dispersed, and seem quite far from experimental values (Young modulus).1. The authors briefly explain how they estimate Young's modulus. In table 1 they also put bulk modulus and glass transition temperature. However, they do not explain how they could estimate these. In particular, Tg is quite a challenge to estimate. It is unfortunate that the authors did not discuss the relation between network structure and these properties, although they have the data.
2. The authors only consider strand percolation. Their systems do not seem to form 3D networks. It seems that most trivalent monomers actually reacted only twice or less. I find this rather interesting and counter-intuitive. A discussion is missing here in my opinion. The final cross-linking density is important missing information. Actually I do not see how the percolation density of table 2 is properly defined in a true 3D network with branches.
3. The very high Young modulus is striking. In soft elastomers, the relaxation time is surely much longer than a few ps (I would rather expect hundreds of ns or more), especially close to the percolation threshold. I suspect that the simulations are much too short for correctly assessing the mechanical moduli (and likely Tg). A way to investigate this is to run a long NVT simulation, computing the time correlations of components of the pressure tensor (fix ave/correlate/long ?) and then the stress relaxation modulus G(t). Since the QM part of the model is not needed for simulations of the mechanics, they should be fast with such small systems.
Author Response
Reviewer 2
Comments and Suggestions for Authors
This manuscript is about the numerical simulation of silicon networks. The authors investigate the relations between system preparation, network structure and mechanical properties. They use a previously published mixed QM-MM molecular model for describing the crosslinking process. The network analysis is described in terms of percolating "strands", loops and free molecules.
The main point of the manuscript is, that percolation of strands are the more likely to correlate to the Young Modulus.
Due to the limited size of the simulation box (QM-MM remains challenging), the mechanical results are quite dispersed, and seem quite far from experimental values (Young modulus).
We point out that our models do not reflect silicone thermosets (which are indeed much softer), but to silicone resins. Comparing our Youngs moduli to those known for silicone resins, we indeed find quite good agreement. This is now described more clearly in the conclusion.
The authors briefly explain how they estimate Young's modulus. In table 1 they also put bulk modulus and glass transition temperature. However, they do not explain how they could estimate these.
In particular, Tg is quite a challenge to estimate. It is unfortunate that the authors did not discuss the relation between network structure and these properties, although they have the data.
Thanks for this hint. We now provide the details as a new paragraph of the methods section:
To assess the glass transition temperatures Tg , we performed heating/cooling runs between 50 K and 500 K, respectively, at rates of 10 K/ns. From this, we analyzed profiles of the system density as functions of temperature. Upon fitting linear asymptotes to the profiles obtained by heating and cooling, the resulting crossing points are taken as estimates for Tg, respectively.
The authors only consider strand percolation. Their systems do not seem to form 3D networks. It seems that most trivalent monomers actually reacted only twice or less. I find this rather interesting and counter-intuitive. A discussion is missing here in my opinion. The final cross-linking density is important missing information. Actually I do not see how the percolation density of table 2 is properly defined in a true 3D network with branches.
No, our systems actually do form 3D networks. Mind that the number of percolating strands was investigated in all 3 directions and was always found to be larger than 1. To avoid misunderstanding, we explicitly now pointed out in the caption of fig. 2 and in the discussion the percolation analysis.
- The very high Young modulus is striking. In soft elastomers, the relaxation time is surely much longer than a few ps (I would rather expect hundreds of ns or more), especially close to the percolation threshold. I suspect that the simulations are much too short for correctly assessing the mechanical moduli (and likely Tg). A way to investigate this is to run a long NVT simulation, computing the time correlations of components of the pressure tensor (fix ave/correlate/long ?) and then the stress relaxation modulus G(t). Since the QM part of the model is not needed for simulations of the mechanics, they should be fast with such small systems.
We point out that our models do not reflect silicone thermosets (which are indeed much softer), but to silicone resins. Comparing our Youngs moduli to those known for silicone resins, we indeed find quite good agreement. This is now described more clearly in the conclusion.
Regarding the relaxation times, please mind that we use an adaptive relaxation strategy to avoid issues of insufficient sampling. Indeed ps scale relaxation runs are only used in the very beginning of the curing process, and individual relaxation runs are applied after each reaction step. For sampling the glass transition temperature and likewise for assessing the elastic moduli, we subjected the finally cured resin models to 100 ns scale runs. The underlying relaxation and sampling periods we carefully tested by splitting our data in two halves and using the deviation from the average as error margins.

Round 2
Reviewer 1 Report
Comments and Suggestions for Authors
Although the author has responded to the questions, no changes have been made in the manuscript. Therefore, the following issues still need to be revised.
- In the abstract section, it is necessary to highlight the reason for conducting this research.
- The author's reply stated that there are currently relevant reports on the use of molecular dynamics to study the curing behavior of organosilicon and predict its performance. The introduction should list relevant previous studies and highlight the innovativeness of this work.
- The table should be in three-line format.
Author Response
Comment1: In the abstract section, it is necessary to highlight the reason for conducting this research
we feel that our abstract well delivers on this purpose. Only small changes were made - as highlighted in yellow.
Comment2: The author's reply stated that there are currently relevant reports on the use of molecular dynamics to study the curing behavior of organosilicon and predict its performance. The introduction should list relevant previous studies and highlight the innovativeness of this work.
Our reply in round pointed out that refs. 8-10 are key to the QM/MM based study of polymer curing. Specific to organosilicon polymers, only ref. 8 is of relevance to our study. We feel that our introduction provides a good balance of needed details and conciseness - we thus wish to keep it this way. The corresponding paragraphs are highlighed in yellow.
Comment3: The table should be in three-line format.
We now also reformatted table 1. Now both tables should be well.
Reviewer 2 Report
Comments and Suggestions for Authors
1) resin vs. thermosets. Could the authors explicit what distinction they consider? Is this a difference of crosslink density or chemical nature? They mention that the Young moduli they obtain are in agreement with experiments on silicone resins. Actually they give just 1 citation, which is about phenyl silicone resin. This is not convincing.
I still think that the crosslink density is important missing information.
2) percolation of "strands". If the polymer actually forms a 3D network, speaking of strands becomes ambiguous. Note that percolation of a single strand across the 3 box faces does not imply that the polymer forms a 3D network (e.g. fig 3b). If the "strands" do form a network with percolating branches, then the number of percolating "strands" (that the authors call percolation "density") depends on the position of the cutting plane across which they are counted. This is not a reliable indicator.
Author Response
comment1: 1) resin vs. thermosets. Could the authors explicit what distinction they consider? Is this a difference of crosslink density or chemical nature? They mention that the Young moduli they obtain are in agreement with experiments on silicone resins. Actually they give just 1 citation, which is about phenyl silicone resin. This is not convincing.
I still think that the crosslink density is important missing information.
In the conclusion section, we mistakingly wrote 'thermoset' instead of 'elastomer'. This is now amended. While silicone elastomers were subjected to abundant experimental studies, we aggree that there is a lack of experimental evidence related to highly crosslinked silicone resins. Actually, the named reference is the only and most applicable study we found. This is now described more clearly.
comment 2) percolation of "strands". If the polymer actually forms a 3D network, speaking of strands becomes ambiguous. Note that percolation of a single strand across the 3 box faces does not imply that the polymer forms a 3D network (e.g. fig 3b). If the "strands" do form a network with percolating branches, then the number of percolating "strands" (that the authors call percolation "density") depends on the position of the cutting plane across which they are counted. This is not a reliable indicator.
No, graph theory can actually manage this issue. We now point this out more clearly in the methods section. Namely, any biassing from the position of the box boundaries is resolved by translating the polymer strands into graphs. We do not use a 'cutting plane' but an identifyier for closed loops through the 3D periodic boundaries. This fully avoids ambiguity.
We thank the reviewer for this discussion - our approach to assessing the percolation density is now described in more detail in the methods section to clarify this point.
Round 3
Reviewer 2 Report
Comments and Suggestions for Authors
I suggest to accept publication in the current form.